# Efficient Near-Infrared Spectrum Detection in Nondestructive Wood Testing via Transfer Network Redesign

**DOI:** 10.3390/s24041245

**Published:** 2024-02-15

**Authors:** Dapeng Jiang, Keqi Wang, Hongbo Li, Yizhuo Zhang

**Affiliations:** 1College of Computer and Control Engineering, Northeast Forestry University, 26 Hexing Rd., Harbin 150040, China; jiangdapeng1992@nefu.edu.cn (D.J.); zdhwkq@163.com (K.W.); 2College of Electrical and Information, Northeast Agricultural University, Harbin 150030, China; lihongbo@neau.edu.cn; 3College of Computer Science and Artificial Intelligence, Changzhou University, 1 Gehu Middle Rd., Changzhou 213164, China

**Keywords:** CNN, domain adaptation, near-infrared detection, nondestructive testing, transfer learning, wood material analysis

## Abstract

This study systematically developed a deep transfer network for near-infrared spectrum detection using convolutional neural network modules as key components. Through meticulous evaluation, specific modules and structures suitable for constructing the near-infrared spectrum detection model were identified, ensuring its effectiveness. This study extensively analyzed the basic network components and explored three unsupervised domain adaptation structures, highlighting their applications in the nondestructive testing of wood. Additionally, five transfer networks were strategically redesigned to substantially enhance their performance. The experimental results showed that the Conditional Domain Adversarial Network and Globalized Loss Optimization Transfer network outperformed the Direct Standardization, Piecewise Direct Standardization, and Spectral Space Transformation models. The coefficients of determination for the Conditional Domain Adversarial Network and Globalized Loss Optimization Transfer network are 82.11% and 83.59%, respectively, with root mean square error prediction values of 12.237 and 11.582, respectively. These achievements represent considerable advancements toward the practical implementation of an efficient and reliable near-infrared spectrum detection system using a deep transfer network.

## 1. Introduction

Near-infrared spectroscopy (NIRS) stands out for its rapid and user-friendly operation, high efficiency, and nondestructive measurement capabilities, which make it a versatile tool for both qualitative and quantitative analyses of fundamental components in various samples as well as for detecting adulteration in samples [1]. Using solid wood spectroscopy as an illustration, the distinctive absorption bands of essential wood components, including cellulose, lignin, and hemicellulose, align closely with the overtones or fundamental vibrations in the near-infrared (NIR) spectrum. Moreover, organic substances exhibit notably weaker absorption of NIR light [2]. Leveraging these characteristics, NIRS has become instrumental in revealing the diverse features of wood samples.

In the realm of analytical techniques, researchers have commonly employed a combination of statistical and computer-based methods, collectively known as chemometrics, to build statistical models for extracting valuable substance information from NIR spectra. Among these techniques, partial least squares (PLS) shows prominence for its widespread application in classifying and modeling NIR spectra [3]. While methods such as backpropagation (BP) neural networks have not shown marked advantages over PLS in classification and prediction, recent years have witnessed a transformative shift. Deep learning models based on NIRS, particularly convolutional neural networks (CNNs), have outperformed traditional PLS models in nondestructive detection [4,5,6,7,8]. This marks a notable advancement in the field, demonstrating the increasing efficacy of deep learning approaches in the context of NIR spectroscopy.

The primary objective of researchers is establishing a model that eliminates the need for repetitive chemical analyses and destructive experiments in the modeling with NIRS [9,10,11]. However, challenges arise when existing research equipment must be replaced or damaged components in older instruments must be repaired, potentially requiring the repetition of various previously conducted calibration experiments [12]. In response to this concern, some researchers have developed calibration methods based on discerning differences between new and old instruments to rectify the model [13,14,15,16,17]. This corrective process, known in the field of chemometrics as calibration transfer (CT) [3], ensures the adaptability of existing models to new instruments. In fact, the concept of calibration transfer extends beyond near-infrared spectroscopy to other analytical chemistry techniques such as Raman spectroscopy and nuclear magnetic resonance [15,18]. We focus solely on near-infrared spectroscopy as a foundation, discussing how calibration transfer models can undergo further modifications.

In the domain of deep learning, transfer learning has found extensive application in areas such as natural language processing and computer vision. However, for researchers and model developers specializing in CT techniques, transfer learning is a relatively recent and evolving area. Many deep learning architectures are not directly applicable to chemometrics. Consequently, current research is predominantly focusing on leveraging networks such as CNNs to establish transfer models through feature sharing [12]. Notably absent from the current research landscape are in-depth explorations of the fundamental concepts of deep transfer learning, including domain transfer, feature transfer, feature representation, and their tailored implementation in chemometrics. Moreover, the ongoing research also lacks a comprehensive analysis of deep neural networks to identify structures that are unsuitable for chemometric applications. Addressing these gaps in understanding is crucial for advancing the field of transfer learning in chemometrics, ensuring that deep learning models are effectively adapted and optimized for the unique challenges posed by NIRS and its calibration transfer requirements.

Among the deep learning scientists familiar with transfer learning techniques, transfer learning is often referred to as domain adaptation technology. As the landscape of domain adaptation evolves, deep transfer networks have taken on diverse and intricate structures. This study focuses on the application of deep neural network models in chemometrics and nondestructive testing. In chemometrics, obtaining labeled data through wet chemical experiments poses challenges, despite the ready availability of nondestructive testing data such as spectra and chromatography-mass spectrometry for samples. Consequently, this study places particular emphasis on unsupervised domain adaptation (unsupervised DA).

The landscape of unsupervised DA has manifested in three primary neural network structures: distance-based methods, optimal transport (OT), and adversarial machine learning. Common and effective unsupervised domain transfer networks often incorporate combinations of these three primary structures along with specialized modules [19,20]. Within the distance-based methods, Gretton et al. proposed maximum mean discrepancies (MMDs), which is a deep transfer network-friendly method that has become widely adopted in various transfer networks for transfer loss [21]. Long et al. expanded on this method by integrating kernel functions into MMD, introducing the MK-MMD method and establishing an early deep transfer network known as the deep adaptation network (DAN) [22]. Zhang et al. proposed MDD, a novel measurement method with rigorous generalization bounds, specifically designed for distribution comparison using asymmetric margin loss and minimax optimization for enhanced training [23]. Within the OT structures, Phan et al. introduced a novel regularization technique within the Wasserstein-based distributional robustness framework, which is termed globalized loss optimization transfer (GLOT) [24]. Li et al. proposed enhanced transport distance (ETD) for unsupervised DA, developing an attention-aware transport distance as the prediction feedback of an iteratively learned classifier to measure domain discrepancy [25]. Within the adversarial machine learning structures, Chen et al. developed the batch spectral penalization (BSP) structure, integrating it with an adversarial layer to form a novel neural network named conditional domain adversarial network (CDAN) [26]. However, similar to deep learning, the fundamental structures in domain adaptation methods have not been systematically analyzed from the perspective of near-infrared spectroscopy mechanisms and chemometrics. To address this issue, this study proposes the following:An exploration of the applicability of the basic structures in deep networks, including convolutional structures, fully connected structures, pooling structures, etc., and domain adaptation network techniques such as maximum mean discrepancies, optimal transport, and adversarial machine learning from the standpoint of near-infrared spectroscopy analysis.Based on the analysis presented in this paper, a modification of five mainstream domain adaptation networks was conducted to enable these networks to meet the requirements of near-infrared spectroscopy calibration transfer models, thereby facilitating their application in near-infrared spectroscopy analysis technology.

This study endeavors to offer a comprehensive exploration of the distinctive features inherent in various neural network structures. We delve into the nuanced characteristics of these structures, elucidating how they can be effectively implemented in the realm of chemometric models. Our analysis extended to investigating the profound impact of NIRS datasets on the architecture of these networks. Furthermore, we aimed to enhance existing models, ensuring their practical applicability in the field of chemometrics. We meticulously selected and scrutinized five transfer learning models. These models were developed utilizing a NIRS dataset, with the mechanical properties of wood serving as the labels. Our objective was to comprehensively assess the accuracy and efficacy of these models. The experimental results unequivocally demonstrated the effectiveness of our approach in the domain of CT, yielding superior performance on the NIRS dataset concerning mechanical wood properties. This verified the viability and potential of our proposed methodology for enhancing and optimizing chemometric models through the integration of transfer learning techniques, particularly within the context of NIR spectroscopy datasets.

## 2. Theories and Methods

### 2.1. Fundamental Structure of Deep Neural Networks

Deep neural networks, encompassing recurrent neural networks (RNNs) and CNNs, are pivotal in various applications [27,28]. RNNs excel in processing sequences, such as in natural language tasks or stock market trend predictions. However, when analyzing NIRS, the linear correlation between spectral bands and organic functional groups suggests a different approach is required. Statistical and machine learning methods can directly model specific spectral bands, yielding precise calibration models without the need to treat NIRS data as inherently sequential. Consequently, employing RNNs for NIRS modeling may be inappropriate. Instead, using CNNs by selecting spectral bands corresponding to characteristic functional groups is deemed more suitable.

In this study, we adopted CNN modules, including convolutional layers, pooling layers, fully connected layers, dropout layers, and activation functions, as the fundamental building blocks for a deep transfer network tailored to NIR spectrum detection. Our evaluation focused on their suitability for our study context. Notably, RNNs are not practically used for modeling, and their accuracy is not comparable to that of fully connected network models. The presented results are theoretical analyses, offering potential directions for future research.

#### 2.1.1. Convolution Layer in Chemometrics and  NIRS

In textbooks and educational materials, deep networks often process input as an infinite-length discrete signal, whether one- or two-dimensional. However, focusing on chemometrics and metabolomics in this study, which involve techniques such spectroscopy and gas chromatography–mass spectrometry, reveals that signals from experiments are finite-length, one-dimensional discrete signals. Recognizing this unique data characteristic, we intentionally narrowed our focus to the domain of one-dimensional discrete signals. Unlike the idealized infinite-length signals in theoretical contexts, practical experiment signals have a defined limited length. Acknowledging this finite nature and aligning research methodology with actual experimental data intricacies, we chose to center our investigation on one-dimensional limited discrete signals *y*.
(1)$$y=[y_{0},y_{1},y_{2},\ldots ,y_{k}]$$
where $y_{k}$ represents the signal amplitude measured in *k* time, and, in chemometrics, it signifies the *k*th spectral band. *y* is a signal with finite amplitudes, and it can be defined as $\max_{j}\left|y_{j}\right|<\infty$, i.e., ${∥y∥}_{1}<\infty$. Assuming the weight function is $w=[w_{0},w_{1},w_{2},\ldots ,w_{k}]$, the convolution result *h* of the one-dimensional discrete signal is given by
(2)$$h\left(k\right)=\sum_{i=1}^{k}y_{i}\times w_{k-i}$$

A digital image is an intricate arrangement of pixels seamlessly combined to create a visual representation containing detailed information such as the contours and shapes of objects. Convolution operations play a pivotal role in the realm of image processing, aiding in tasks such as feature extraction and other critical areas. Prior to the advent of CNNs, these convolution operations were fundamental in image processing, finding widespread application in edge detection, filtering, and texture feature extraction.

Acquarelli et al. achieved a milestone in introducing CNNs to NIRS, constructing a deep learning network with 1×n convolutional layers [29]. Following their pioneering work, a considerable number of studies have embraced similar approaches in combining NIRS with deep learning since 2017–2018 [4,30,31,32,33].

Despite these advancements, notably, the modeling logic of NIRS substantially diverges from that of image processing. Historically, the impact of convolution on NIR spectra has been a focal point of attention, as evident in early works such as P.B. Tooke’s research [34,35]. In their articles, the authors express their perspective using Figure 1.

In Figure 1a, the original signal of the NIR/IR spectrum comprises multiple absorption peaks denoted as *x*. The NIR/IR spectrum is viewed as a response *s* formed by the “interweaving” of these original signals. The figure cited from Tooke’s article illustrates how the interwoven NIR spectrum can be separated into multiple absorption peaks. The objective of deconvolution is to deduce the original signal *x* from the response and impulse response *s*. Figure 1b,c describe this process. In Figure 1c, a single absorption peak represents a characteristic functional group of the tested substance. For this study, this may correspond to the C–H or O–H groups associated with cellulose.

While Fourier transform has traditionally been a primary algorithm for deconvolution, the past decade has seen a shift with the emergence of the wavelet transform. Despite this, the fundamental idea of decoupling signals through deconvolution operations has gained widespread acceptance in the optics field.

NIR spectra result from the coupling of multiple bands, each corresponding to specific chemical bonds of the measured substance. The objectives of chemometrics, wavelet transforms, and multivariate statistical regression are the establishment of a one-to-one correspondence between feature bands and labels. For feature bands requiring decoupling, the importance of deconvolution methods, particularly inverse convolution, cannot be overstated. Continuing convolution on spectra would only complicate feature bands, increasing their intricacy and overlap.

Specifically, the main absorption bands correspond one-to-one with the chemical bonds in the spectrum in NIR. The convolution operation, however, disrupts this correspondence unless the convolution kernel adheres to a specific form, such as $w\left(a\right)=\left[0,\ldots ,0,1,0,\ldots ,0\right]$. Without this specific form, drawing conclusions such as the following—the “application of CNNVS to data which was not preprocessed as well as to preprocessed data yielded superior accuracy performance compared to PLS-LDA and Logistic Regression”—as suggested by Acquarelli and others, requires more rigorous demonstration.

According to Acquarelli et al., the convolutional layers in NIR spectra address issues such as noise and baseline drift. In the nearly nonexistent landscape of NIR-spectrum-preprocessing algorithms in the 1990s, one-dimensional convolutional networks were considered innovative and capable of addressing some noise and baseline drift issues. Now, the S-G smoothing algorithm and first-order derivative methods offer tailored solutions for specific challenges in spectral modeling, effectively mitigating the effects of noise and drift in spectral data. In contrast, researchers have opted for CNN as a black-box model, achieving preprocessing through model training and data fitting. This approach seems less convincing.

To elaborate further, let us consider an example in the field of partial differential equations. In this field, some equations lack analytical solutions, and scholars have obtained numerical solutions using various approximation and fitting algorithms, comparing computational speed and accuracy. However, once scholars obtain a numerical solution to the partial differential equation, related work in the field typically concludes. Similarly, if scholars in the field of near-infrared analysis have not proposed the Savitzky–Golay smoothing algorithm or first-order derivative methods, using convolutional networks to model and suppress noise and baseline drift in spectra would be considered a remarkable achievement. but in the presence of the Savitzky–Golay smoothing algorithm and first-order derivative methods, avoiding the use of convolutional neural networks for preprocessing spectral data might be a better choice.

Regarding preprocessing techniques such as the S-G convolutional algorithm that deviate from the established logic of near-infrared spectroscopy analysis, it is advisable to exercise caution in incorporating similar concepts into expansive models, which may result in opaque, black-box solutions. Instead, a prudent approach involves the selective application of such methods. Relying on the expertise of scientists, an evaluation should be conducted to determine whether the spectrum exhibits any discernible enhancement after preprocessing. If distortions are identified in the spectrum arising from the S-G method, it is prudent to abandon it in favor of alternative denoising strategies.

Moreover, although improvements in performance have been acheivced for many CNN models, this advancement is plausibly linked to the “No Free Lunch” effect. Nevertheless, we must consciously choose methodologies that are more grounded and rational.

In summary, drawing from my limited knowledge of NIRS and deep learning, I provide a concise overview:A straightforward multivariate linear connection exists between the concentrations of the components and the group characteristics observed in NIR spectra. In such a situation, indiscriminate convolution operations only disrupt the identification of characteristic spectral bands.The NIR spectrum results from the amalgamation of absorption peaks, and the deconvolution method, employing Fourier transform, can effectively disentangle the spectrum into individual absorption peaks. Conversely, the convolution operation lacks both physical and practical meaning within this particular context.

#### 2.1.2. Fully Connected Layer in Chemometrics and  NIRS

The fully connected layer is a foundational element within diverse neural network architectures, spanning from simpler feedforward neural networks and BP neural networks to more intricate deep learning structures. In the realm of the fully connected layer, each layer acts as the input for the subsequent one. Assuming the input vector of the preceding layer is denoted as $y_{dense}$, the output is mathematically represented as
(3)$$h=g\left(W\cdot y_{\mathrm{dense}}+b\right)$$

As shown in Equation (Equation 3), the fully connected layer embodies a distinctive form of the generalized linear model (GLM). In statistical terms, a GLM is a remarkably adaptable statistical model. This model posits the distribution function of random variables measured using chemical instruments, which is modeled by a link function and is correlated with experimental results and labels. Delving into the pertinent literature on NIRS calibration modeling techniques unveils that each wavelength point in the NIR spectra encapsulates information from multiple components. The integration of modern mathematical multivariate statistical algorithms can establish a robust relationship model between sample NIR spectra and sample labels. The generalized linear model has emerged as an exceptionally effective multivariate statistical algorithm. Reflecting on the developmental trajectory of NIRS modeling techniques over the past two decades, neural networks have emerged as pivotal contributors.

For instance, Zhao et al. used NIR spectra and BP neural network algorithms, along with ANN algorithms, to monitor the content of free fatty acids in fried oil samples [36]. Zeng et al., in a review, proposed that the BP-ANN model can be used to establish an NIRS identification model for apple varieties [37]. Notably, in 2016, Chen et al. achieved CT using extreme learning machines (ELM) with three spectral datasets of corn and tobacco [38]. The authors briefly analyzed the rationale behind the success of the ELM autoencoder in achieving model transfer for NIR spectra.

In essence, for the integration of deep network technology into NIRS and the attainment of transfer learning, the fully connected layer is indispensable. This study aimed to construct a transfer network grounded in the fully connected layer. However, in light of the earlier study by Chen et al. [38], which did not leverage mainstream deep learning libraries such as PyTorch 1.2.0 or TensorFlow 1.9.0, this study aimed to overcome this limitation by employing the PyTorch library to establish a NIRS transfer model.

#### 2.1.3. Maxpool Layer in Chemometrics and  NIRS

Pooling is a vital technique in both machine learning and image processing, offering a condensed representation of input data via selecting crucial local features such as maxima, minima, or averages. The primary functions of pooling are twofold: first, it provides feature invariance, allowing the objects in an image to be recognized even after pooling. Techniques such as local maxima and minima pooling facilitate the capture of essential edge features by neural networks. Second, pooling contributes to feature dimensionality reduction, aiming to eliminate redundant information, extract relevant image features, reduce model complexity, enhance operational speed, and decrease the likelihood of overfitting.

However, in the context of nondestructive testing using NIRS, the length of one-dimensional nondestructive testing signals is typically within 1000. When employing fully connected layers to extract signal features, assuming the feature layer is also 1000, the floating-point operations per second (FLOPS) would be $10^{6}$, differing by at least 1000 orders of magnitude from the computational intensity of mainstream deep networks [39,40]. Consequently, chemometrics models do not require increased model operational speed. Moreover, the data length collected by NIR spectrometers remains consistent, obviating the necessity of pooling operations to standardize data dimensions.

Furthermore, in image processing, the edges and textures of data images can be extracted through neighborhood max–min sampling (early developments in pooling technology). However, NIR spectra often feature overlapping absorption peaks, presenting a challenge in determining the crucial band in a given neighborhood. Nevertheless, considering the concern with model overfitting, mechanisms akin to pooling layers should still be incorporated into deep networks.

In conclusion, this study opted for a dropout layer as a viable substitute for the pooling layer, constructing a feature extraction module with the following structure: “fully connected layer—dropout layer—fully connected layer—dropout layer”. This approach aims to strike a balance between feature extraction and mitigating overfitting in the context of NIRS analysis.

#### 2.1.4. Activation Function in Chemometrics and  NIRS

Concerning the activation function, the deep networks built upon NIR spectra do not necessitate an intricate model structure with numerous layers. Consequently, they are notably less complex than image processing algorithms, thereby mitigating the issues known as vanishing and exploding gradients in our designed network. Within this context, the impact of different activation functions on network accuracy is nearly negligible. Following a meticulous comparison of three activation functions—ReLU, sigmoid, and tanh—this study opted for a combination of ReLU and sigmoid as the activation functions for constructing our foundational network.

#### 2.1.5. Summary

With these decisions in mind, we provisionally determined the specific modules and structures to be employed in constructing the NIR spectrum detection model. This determination stemmed from a comprehensive analysis and the careful consideration of fundamental structures such as fully connected, pooling, and convolutional layers. Such a thorough examination provided clear guidance, ensuring the effectiveness and performance of the model. Building upon these foundational choices, our next steps involved delving deeper into research and optimization within the domain adaptation branch. This ensured the ultimate development of an efficient and reliable NIR spectrum detection system that seamlessly aligns with practical application requirements. This iterative process underscores our commitment to refining and enhancing the model to meet the specific demands of NIR spectrum analysis.

### 2.2. Unsupervised Transfer Learning

In this section of the study, we singled out five classical transfer networks and provided concise descriptions of their transfer architectures. Building upon our previous insights into fundamental structures such as fully connected, pooling, and convolutional layers, we modified these networks to better align them with the specific demands of modeling NIR spectra. This strategic step was geared toward refining the networks and optimizing their architecture to enhance adaptability to the unique characteristics of NIR spectra. By redesigning these classical transfer networks to fit the nuances of the NIR calibration model, we aimed to fortify the model’s performance and applicability and bolster the model’s generalizability, ensuring it excels in accuracy and stability across diverse practical applications, thereby fostering outstanding outcomes in real-world scenarios.

#### 2.2.1. Multiple Kernel Variant of MMD

The MMD algorithm, introduced by Gretton et al., is a robust measure for assessing whether two samples originate from distinct distributions, making it an algorithm choice in domain adaptation [21]. Its versatility and effectiveness in chemometrics are particularly noteworthy, addressing crucial questions such as the compatibility of data collected in different laboratories or the comparability of microarray data from diverse tissue types. As highlighted by Gretton et al., MMD finds application in scenarios with interest in comparing microarray data from identical tissue types measured at different laboratories, aiming to discern whether the data can be jointly analyzed or if differences in experimental procedures introduced systematic variations in data distributions [21].

Whereas MMD has found widespread use in subsequent image feature processing, it has also achieved substantial inroads into the domain of deep learning networks. The fundamental premise of the MMD method lies in the notion that if all moments of two random variables are equal, the two distributions are deemed identical. Conversely, if the distributions differ, the objective is to identify the moment that maximizes the difference between them, serving as a metric for measuring dissimilarity. This loss function proves invaluable in quantifying the distance between two distinct yet related random variable distributions. Its application extends beyond traditional statistical measures, making it a powerful tool in the realm of deep learning, especially in scenarios where understanding and mitigating distributional differences are paramount [21].
(4)$$MMD\left(s,t\right)={∥E_{s}\left[\phi \left(y^{s}\right)\right]-E_{t}\left[\phi \left(y^{t}\right)\right]∥}^{2}$$

$\phi \left(y\right)$ represents a mapping function that maps data features into a high-dimensional space; *s* and *t* denote the distributions of the source domain and target domain, respectively.

This study found inspiration in the DAN architecture proposed by scholars such as Mingsheng Long, which is rooted in the MMD distance algorithm. To tailor the DAN network to our specific needs, we implemented personalized and customized adjustments. In the final iteration of the deep network architecture, we departed from the conventional convolutional layer structure, opting instead for three fully connected layers to directly extract features from the spectra.

In these three fully connected layers, the first and third layers represent the private components of the neural network for the source and target domains, respectively. This design ensures the independence of each domain. The second layer functions as the shared component, facilitating the learning of features shared between the domains. During the training process, we initially fixed the first and third fully connected layers corresponding to the source domain (target domain) and trained using target domain (source domain) data. Subsequently, we jointly trained the shared part, using both source and target domain data to fine-tune the model. The final process structure is depicted in Figure 2.

In Figure 2, “input_t” and “output_t” represent the target domain data, and “input_s” and “output_s” represent the source domain data. “Squeeze”, “Unsqueeze”, “slice”, and “concat” correspond to the PyTorch API instructions used for slicing and concatenation when handling torch data. Typically designed for classification tasks, DANs underwent crucial modifications in this study to adapt them for regression prediction. These alterations included replacing the original softmax activation function with a sigmoid activation function and substituting the cross-entropy loss function with mean squared error (MSE). These adjustments were applied to better align the network with the requirements of regression prediction tasks. This personalized customization not only increases the adaptability of the model but also provides a more robust tool for the regression analysis of spectral data.

#### 2.2.2. Conditional Domain Adversarial Network

The authors of the CDAN develoepd a principled framework that conditions the adversarial adaptation models on the discriminative information conveyed in the classifier predictions [26,41]. They leverage this characteristic by using the discriminator of a GAN as a domain adaptation component to extract common features from the source and target domain data. When the discriminator fails to distinguish the origin domain of the input data, the extracted features are shared between the source and target domain data. Based on this, they proposed a transfer strategy based on conditional domain adversarial learning. These researchers improved adversarial domain adaptation methods, defining the CDAN as a minimax optimization problem with two competing error terms:(1)Let *G* be the classifier of the source domain deep network and L(·,·) be the cross-entropy loss. To ensure a lower source domain error risk, we seek the deep network solution that minimizes the expected E(G).(2)Let *D* be the discriminator for the cross-source and target domains. Set f=F(y) and g=G(y), corresponding to the classifier result and network feature representation, respectively; and D(f;g) is the discriminator result for *g* and *f*. Then, E(D,G) can be represented as a joint distribution across the source and target domains [41].
(5)$$E\left(G\right)=E_{\left(x_{i}^{s},y_{i}^{s}∼D_{s}\right)}L\left(G\left(y_{i}^{s}\right),{label}_{i}^{s}\right)$$
(6)$$E\left(D,G\right)=-E_{x_{i}^{s}∼D_{s}}\log\left[D\left(f_{i}^{s},g_{i}^{s}\right)\right]-E_{x_{j}^{t}∼D_{t}}\log\left[D\left(f_{j}^{t},g_{j}^{t}\right)\right]$$

Organizing Formulas (Equation 5) and (Equation 6), the minimax optimization of the CDAN is [41]:(7)$$\min_{G}E\left(G\right)-\lambda E\left(D,G\right)$$
(8)$$\min_{D}E\left(D,G\right)$$

Finally, the CDAN network divides samples based on the difficulty of classification, which are measured using the sample’s entropy [41]:(9)$$H\left(g\right)=-\sum_{c=1}^{C}g_{c}\log g_{c}$$

The larger the entropy, the closer to the boundary point, the more challenging the sample classification, and the smaller the weight. CDAN is only suitable for classifier model discrimination. CDAN measures the difficulty of sample classification based on the sample’s entropy, whereas we modeled regression prediction problems. Entropy cannot be used to measure the difficulty of sample regression prediction problems. Therefore, this study assumed that two sets of labels for NIR spectra that are close are similar in specific bands related to the labels. Based on this assumption, we made some adjustments to the CDAN:Use MSE instead of cross-entropy loss.Propose a new weight assignment formula based on the histogram distribution function constructed from the labeled data in the dataset.

Assuming that the label range of the wood spectral dataset is $label\in [R_{n},R_{m}]$, this range is uniformly divided into *N* segments, and $r_{k}$ represents the *k*th segment, which can be expressed as
(10)$$r_{k}\in \left[R_{n}+\frac{k(R_{m}-R_{n})}{N},R_{n}+\frac{\left(k+1\right)\left(R_{m}-R_{n}\right)}{N}\right]$$

The probability distribution function can be represented as
(11)$$p\left(r_{k}\right)=\frac{n_{k}}{num_{label}}$$

Here, $n_{k}$ is the number of labels in $r_{k}$, and $num_{label}$ represents the total number of labels in the spectral dataset.

Based on the assumption, if the amount of labeled data is larger in a certain interval, there are more NIR spectral data in this label interval, and the relationship between spectral bands and labels is easier to determine by the network. Therefore, the labels corresponding to the spectra in this interval have a higher weight. Assuming *g* is the output of the CDAN, the weight function is represented as
(12)$$w\left(g\right)=\frac{p\left(r_{k}\right)-p_{\min}}{p_{\max}-p_{\min}}\;g\in r_{k}$$

The weighted discriminator can be represented as
(13)$$E^{\prime }\left(D,G\right)=-E_{x_{i}^{s}∼D_{s}}w\left(g\right)\log\left[D\left(f_{i}^{s},g_{i}^{s}\right)\right]-E_{x_{j}^{t}∼D_{t}}w\left(g\right)\log\left[D\left(f_{j}^{t},g_{j}^{t}\right)\right]$$

In terms of the dataset source and target domain feature extraction process, the five network structures are basically the same. We also abandoned the convolutional layer structure and adopted a three-layer fully connected layer structure to directly extract features from the spectrum. The first and third layers within this structure symbolize the private components of the source and target domains within the neural network, respectively, emphasizing the independence of each domain. The second layer functions as a common layer for learning shared features between the domains. The first layer’s outcome serves as the feature representation *f*, whereas the third layer’s output represents the discriminator result *g*, collectively constituting the CDAN branch network.

During the training process, first, the CDAN branch network is trained with source and target domain data. Subsequently, the first and third fully connected layers, corresponding to the source domain (target domain), are fixed and trained with target domain (source domain) data. Finally, the common layer is trained using both source and target domain data, fine tuning the model for enhanced performance. For clarity on the improvements we applied to the CDAN, Figure 3 illustrates the CDAN adversarial layer network structure only. In this figure, “gemm” denotes the matrix multiplication between spectral features and weights, while “relu” and “sigmoid” represent activation functions. This figure depicts the branch network.

We adjusted and optimized the CDAN, making the CDAN network more suitable for the establishment and application of the NIR spectral CT model. These adjustments included several steps. First, the loss function L(·,·) of the CDAN was reconstructed, replacing the traditional cross-entropy loss with MSE. Based on this, the weighted discriminator of the network was reconstructed according to our designed weight assignment shown in Formula (Equation 12). Through this series of improvements, we achieved the application of the CDAN in the field of NIR spectral CT, substantially progressing the research in the field of NIR spectral calibration.

#### 2.2.3. Margin Disparity Discrepancy Method

The MDD method improves the marginal loss function in the transfer network, which is essentially viewed as a variant of the MMD. This model constructs a distance function between the source and the target domain feature representations. The model formulation can be expressed as [23]:(14)$$\min_{f}E\left(\hat{P}\right)+\eta D_{\gamma }\left(\hat{P},\hat{Q}\right)$$
(15)$$\min_{f\prime }D_{\gamma }\left(\hat{P},\hat{Q}\right)$$
where $\hat{P}$ represents the source domain, $\hat{Q}$ represents the target domain, and *D* is the distance from the source domain to the target domain. The MDD method is built on the min–max problem paradigm of GAN-generated networks, confusing the feature representations of the source and target domains to test whether the discriminative network can recognize the differences between the source and target domain features. The generative and discriminative networks engage in mutual adversarial training, making it challenging for the discriminative network to determine differences in the feature representations between the source and target domains.

The MDD network’s training process structure is similar to that of the DAN. To highlight the improvements we achieved with the MDD network, only the structure diagram of the minimax optimization adversarial layer is shown in Figure 4. In this diagram, “sub” represents matrix subtraction.

In summary, this study adopted a strategy similar for the first two networks, which involved discarding convolutional and pooling layers and directly using fully connected layers to extract features from the spectra, constructing both the source and target domains. Additionally, to facilitate effective feature transfer between the source and target domains, we introduced the MDD method. This method optimizes feature distributions during network training, achieving the goal of sharing knowledge between different domains. Through the design and adjustment of the MDD network, we not only accomplished feature learning and transfer between the source and target domains but also simplified the network structure, increasing computational efficiency. This method is an effective solution for cross-domain tasks involving spectral data.

#### 2.2.4. Enhanced Transport Distance for Unsupervised DA

In the Introduction, we mentioned another branch of domain adaptation algorithms based on the OT distance in transfer learning [42]. OT is a mathematical problem that was proposed by Gaspard Monge in 1781, aiming to analyze the similarity between two distributions [43]. The Wasserstein distance or Kantorovich–Rubinstein metric is a distance function defined between probability distributions on a given metric space. It was named after Leonid Vaseršteĭn. The Wasserstein distance can be used to achieve a transformative distance from the source to the target domain distribution function. In transfer learning based on OT, the goal is to find the optimal solution for this transformative distance and use it for backpropagation to optimize neural network weights.

The transfer network constructed based on ETD consists of three parts: a feature extractor network f(·), a classification network η(·), and the Kantorovich potential network g(·). An attention module, constructed with a fully connected network, is introduced to balance the computation of the transport distance.

For most modules in this network, including the feature extraction module and whether the attention module is suitable for NIR spectral modeling techniques, detailed judgments were outlined in the previous sections. The Kantorovich potential network is also composed of fully connected layers, to which we made some adjustments to the optimization goals of this module. Therefore, in this study, the classification network was replaced with a regression network, and the predictive accuracy of the ETD network was modeled and tested. The training process of the ETD network is similar to that of the CDAN, so is not reiterated here. Figure 5 shows the network structure of the Kantorovich potential layer in the ETD. In this diagram, “matmul” represents the matrix multiplication between the spectral feature tensor and the weight tensor, while “transpose” denotes matrix transposition.

#### 2.2.5. Global–Local Regularization via Distributional Robustness

Similar to the ETD network, the GLOT network also adopts the Wasserstein distance: researchers introduced the Stein variational gradient descent algorithm, constructed an OT distance structure, and proposed a new regularization method. The GLOT algorithm includes a semisupervised learning module, a domain adaptation module, and an adversarial machine learning module. Detailed judgments were outlined in the previous sections for modules other than semisupervised learning. For semisupervised learning, the GLOT algorithm constructs decision boundaries in high-dimensional space for spectral data corresponding to different labels and establishes a semisupervised learning model. This method of constructing decision boundaries and continuously optimizing them can be directly applied to NIR spectral modeling. The training process of the GLOT network is similar to that of CDAN and ETD and is thus not reiterated here. Figure 6 shows the network structure of the Kantorovich potential layer in the GLOT.

## 3. Results and Discussion

### 3.1. Experimental Analysis of NIRS Datasets

In this study, we employed a dataset of NIR spectra collected from a series of solid wood boards to assess the performance of the five transfer network models we developed: DAN, CDAN, MDD, ETD, and GLOT. The aim was to evaluate whether these transfer models could surpass the accuracy of traditional models such as PDS and DS. The spectral curves of the NIRS dataset are depicted in Table 1. The spectral bandwidth and shape of the spectra of the same type of solid wood boards collected using different spectrometers exhibit substantial variations. This poses a considerable challenge for spectral CT algorithms.

The dataset consisted of the NIR spectroscopy data and reference values for tensile strength in solid wood panels samples. The NIR spectroscopy data comprised spectral data from 466 samples of solid wood boards, which were measured using different NIR spectrometers. The NIRquest512 spectrometer, produced by Ocean Optics in the United States, covers a wavelength range from 900 to 1700 nm with a resolution of 1.56 nm (512 channels in total). Meanwhile, the NIR-NT-spectrometer-OEM system, manufactured by INSION GmbH in Germany, also captures spectra within the same wavelength range but with a wider interval of 6.83 nm (117 channels in total). reference values for the tensile strength in solid wood panels samples were determined using official analysis methods. On solid wood panel samples, the tensile strength parallel to the grain of the wood (GB/T 1938-2009) [44] was used to determine tensile strength, which was proposed by the Standardization Administration of China. The data and the entire set of programs can be accessed on the Data Availability section.

These calibration models used for data processing were implemented in the Python programming language, using the third-party library PyTorch. The samples were divided into calibration and test sets. The calibration set was used to train the transfer network models, whereas the test set was employed to assess the performance of the CT methods. The sample was split using the random_split function from the torch.utils.data library, where 70% of the samples were allocated to the training set and 30% of the samples were allocated to the test set according to the results in Table 2. For the NIRS dataset used in this study, the basic parameter settings for the five transfer network models were as follows:Optimizer:-RMSProp and Adam optimizers were chosen for the deep networks.-RMSProp optimizer parameter momentum was set to 0.9.-Learning rate was set to 0.001 for both optimizers.-Adam optimizer parameter momentum was set to 0.9.-Weight decay for Adam optimizer was set to 0.01.Activation Function:-The activation function selected was sigmoid.Transfer Loss Weight:-The weight for the transfer loss was set to 0.1.

### 3.2. Comparison and Discussion for Transfer Network Model

In evaluating model quality, a fundamental contradiction exists between deep learning and chemometrics researchers. For the majority of chemometric researchers assessing model quality, the commonly used parameters include confidence intervals, uncertainty measurements, and variability measurements. These parameters, originating from statistics, mostly rely on the assumption that random variables must follow a specific analyzable distribution (independent and identically distributed (i.i.d.)). For instance, from the perspective of random samples, a confidence interval is defined as follows: a random variable χ follows a distribution F, assuming θ is the parameter of distribution F. Random variables are independently sampled *n* times, resulting in a random sample $\left\{X_{1},\ldots ,X_{n}\right\}$. If $u(X_{1},\ldots ,X_{n})$ and $v(X_{1},\ldots ,X_{n})$ such that
(16)$$P(\theta \in \left(u\left(X_{1},\ldots ,X_{n}\right),v\left(X_{1},\ldots ,X_{n}\right)\right))=1-\alpha$$
then $\left(u\left(X_{1},\ldots ,X_{n}\right),v\left(X_{1},\ldots ,X_{n}\right)\right)$ is considered a 1−α confidence interval for estimating the parameter θ.

Whereas the field of machine learning inherits some perspectives from statistics and regression analysis, such as the generalized linear regression model based on statistics and regression analysis, its models do not all assume random variables following analyzable specific distributions. The assumption of independent and identically distributed variables is only present in a small subset of models. Machine learning models are more influenced by optimization theory, such as the sequential minimal optimization algorithm in support vector machine models and information entropy and information gain in random forests. These models only require a set of *d*-dimensional random variables χ in an *n*-dimensional sample space. Due to the questionable analyzability of the distributions in these models, calculating confidence intervals for machine learning models becomes a complex and challenging task, requiring a combination of functional and approximation techniques. Kernel-based support vector machines still have a computable, complex distribution function [45,46] where confidence intervals can be calculated using methods such as Laplace approximation. Random forests lack an analyzable distribution, and the confidence space [47] requires separate calculation through scientific research.

To address this issue, Leslie Valiant proposed probably approximately correct (PAC) learning in 1984 [48,49]. This concept was introduced on the rigorous mathematical foundation of regression models, and PAC theory is no longer confined to a specific analyzable distribution. It provides a machine learning mathematical analysis framework with weaker model assumptions and higher model uncertainty. PAC theory establishes the upper bound of learning algorithm errors based on the generalization error output by the learning algorithm. Additionally, PAC learnable problems have been studied with respect to the number of training samples *n*, which has a theoretical boundary—if *n* satisfies certain conditions, then machine learning can find the optimal solution under an expected generalization error and significance level.

With the advance of deep learning, a problem emerges: the complex network structures result in the majority of high-performance deep models being akin to black-box structures. Researchers of deep networks understand only the inputs and outputs, lacking knowledge as to why these structures exhibit superior performance, which are referred to by deep learning researchers as the “black-box problem”. The PAC analysis framework also prevents the comprehensive and in-depth analysis on networks with numerous layers. These deep networks have thus been controversial in the field of machine learning. Due to the focus of deep networks on inputs and outputs, researchers have predominantly used metrics such as accuracy, F1 score, coefficient of determination (R-squared, R), root mean square error (RMSE), root mean square error of cross-validation (RMSECV), root mean square error of prediction (RMSEP), etc., to interpret model quality. For our purposes, we set aside these controversies and chose R, RMSE, RMSECV, and RMSEP as metrics to measure the performance of the five models.

The relationship curves between training iterations and model accuracy, obtained via training the dataset using different transfer network models, are shown in Figure 7. These figures show that as the number of iterations increases, the loss value decreases and stabilizes. A substantial difference was noted between the loss values and the R and RMSE values. The loss value was considered a reference, and the specific model performance was still determined by the R and RMSE values.

After completing model training, to evaluate the influence of the subset sample size on different calibration methods, ten training sets were created based on the proportion of training set samples to the total dataset. These proportions were 25%, 30%, 35%, 40%, 45%, 50%, 55%, 60%, 65%, and 70%. The findings revealed that the five deep network models considered the influence of iteration time and training set size on the model. Notably, GLOT demonstrated the best predictive performance.

The increase in performance for the transfer learning models was more evident (see Table 2). This was primarily attributed to the fact that in the presence of limited training samples, deep transfer network models encounter substantial challenges in learning spectral features through the adversarial network layer. Consequently, the learning of adversarial network parameters becomes more intricate, resulting in a decrease in accuracy for models such as CDAN, ETD, and GLOT. As the proportion of the training set increases, the performance of domain adaptation models with an adversarial layer structure, such as CDAN, ETD, and GLOT, surpasses that of DAN, MDD, and other models. This is mainly due to the increase in accuracy of the adversarial network layer weights with an increase in the number of training samples. In most cases, deep network models such as ETD and GLOT outperformed DAN and MDD. When the proportion of the training set was small, the DAN model performed better.

In this study, the primary focus was addressing the challenge of the low accuracy and suboptimal model performance observed in model transfer methods, such as PDS and PD. The domain adaptation theory within transfer learning introduces an innovative perspective by considering the main spectrometer as the target domain and the spectrometer as the source domain. The process of transferring the model from the spectrometer to the main spectrometer mirrors the migration of a machine learning model from the source to the target domain. Numerous researchers have proposed diverse architectures, including distance-based methods, optimal transport, and adversarial machine learning techniques, which were previously mentioned and applied in this study to near-infrared spectroscopy technology. However, these architectures are still based on empirical and empirical evidence rather than having a certain convergence in mathematical analysis. Our experimental findings revealed that after a sufficient number of model iterations, the accuracy of transfer models is significantly higher than that of PDS, PD, and other model transfer methods. Notably, regression models have real analysis as their mathematical analysis foundation; machine learning models rely on PAC theory as their mathematical analysis foundation; whereas transfer learning lacks a comprehensive mathematical analysis framework.

Furthermore, the efficacy of classical CT methods for NIR spectra is influenced by the size of the transfer sample. Taking DS, PDS, and SST methods as examples, overall, the performance of these methods improved with an increase in the transfer sample size (Figure 8). Limited data size, such as 30% transfer samples, markedly reduced the testing accuracy for specific subordinate instruments. Among these methods, the SST method achieved acceptable accuracy with 50.0% of all data and experienced a slight improvement when using 70% of all data. Even with these enhancements, the SST method still falls short of reaching the upper accuracy limit observed in models such as CDAN, ETD, and GLOT (Table 3).

Compared to classical spectral CT methods, the GLOT method and the CDAN method significantly improve transfer reliability under different data partitions of subordinate instruments (Table 3). The GLOT method produces the highest average R with the smallest interquartile range and extreme values (Figure 9). The CDAN method also outperforms classical spectral CT methods, demonstrating superior algorithmic performance.

## 4. Conclusions

This study addressed the challenges in obtaining NIR spectral annotation data by introducing five transfer learning methods to increase classification accuracy and reduce the dependency on extensively labeled datasets. We also comprehensively analyzed different layers within deep neural networks in the context of NIR spectral research, exploring domain adaptation algorithms, including MMD, domain adversarial networks, and the Kantorovich potential layer.

In contrast to prevalent CT methods in current NIR spectral technology (DS, PDS, and SST algorithms), the proposed methods exhibit increased accuracy and scalability, especially in large-scale NIR spectral experiments with various manufacturers.

However, the integration of deep learning into chemometrics poses a serious challenge due to the lack of a stable mathematical analysis framework, which prevents researchers from establishing a definite expectation for the stability of deep neural network models. To address the low accuracy of traditional transfer calibration models, we implemented a conscious trade-off by abandoning classical chemometrics model evaluation metrics such as confidence intervals and adopting deep learning model evaluation metrics, achieving substantial improvements in model performance. The hope is that future researchers will dedicate efforts to overcoming this predicament, establishing new mathematical analysis frameworks to increase our understanding fo the stability and interpretability of deep learning models, enabling their reliable application across various domains.

## Figures and Tables

**Figure 1 sensors-24-01245-f001:**
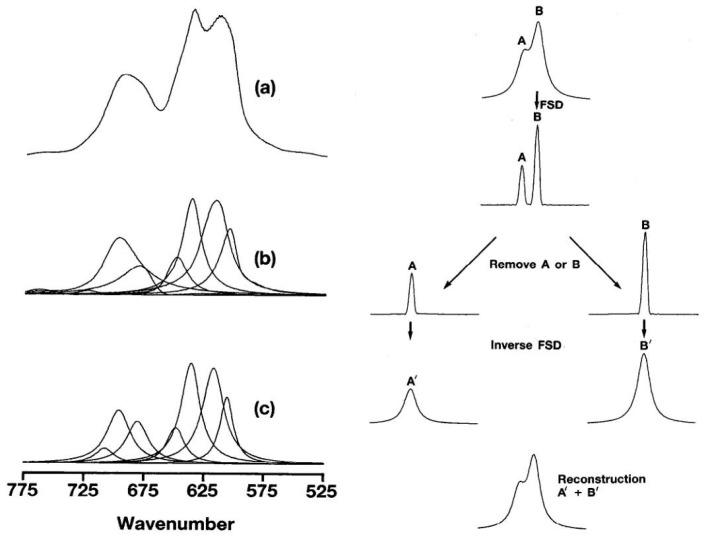
Analysis of the bond of C–Cl stretching region of poly: (**a**) original data (**b**) components by Fourier band isolation, and (**c**) components by curve-fitting (adapted from Figures 7 and 9 in P.B. Tooke’s research [34]).

**Figure 2 sensors-24-01245-f002:**
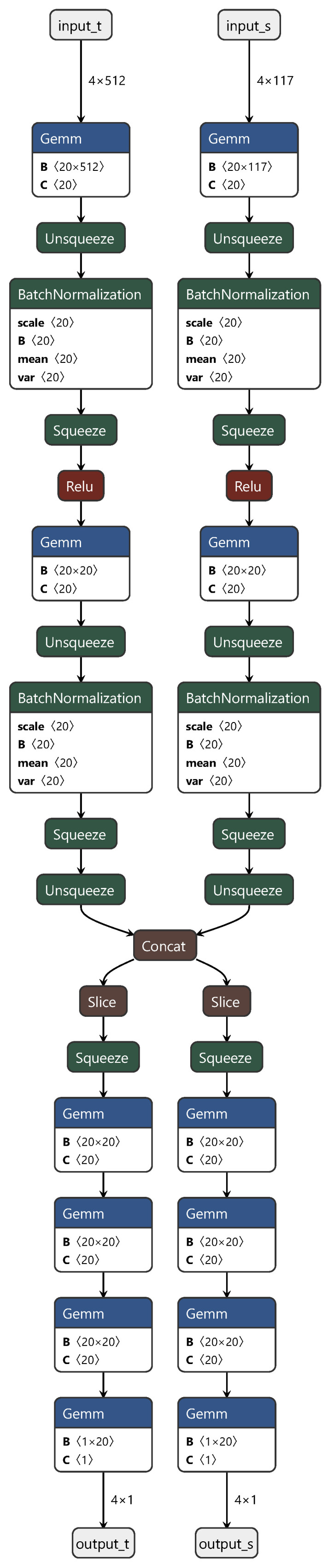
Diagram of the DAN structure.

**Figure 3 sensors-24-01245-f003:**
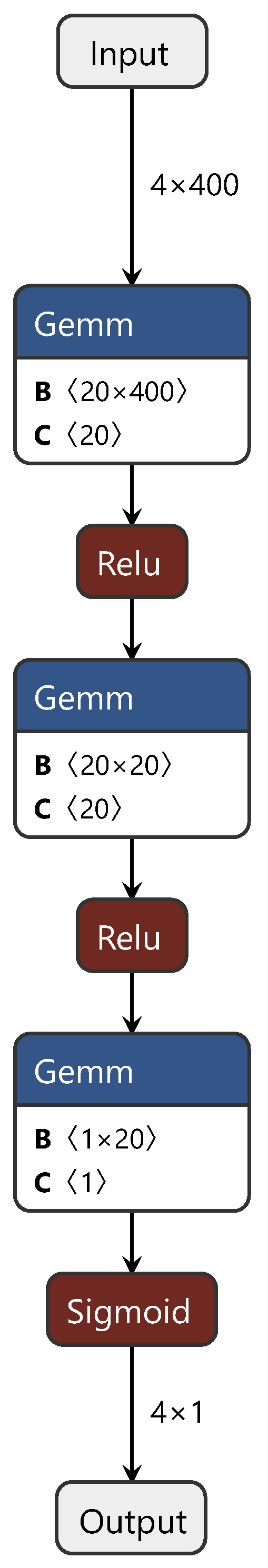
Diagram of the CDAN adversarial layer structure.

**Figure 4 sensors-24-01245-f004:**
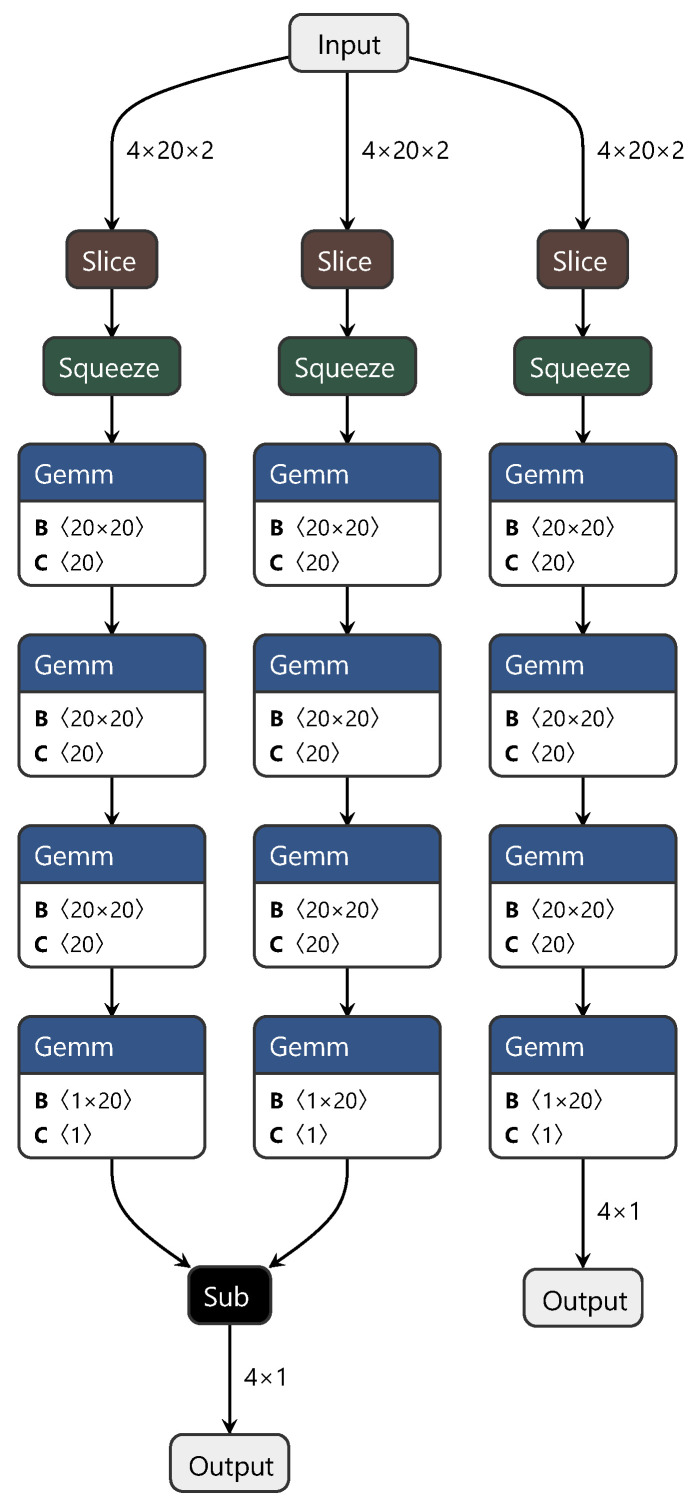
Diagram of the minimax optimization adversarial layer structure.

**Figure 5 sensors-24-01245-f005:**
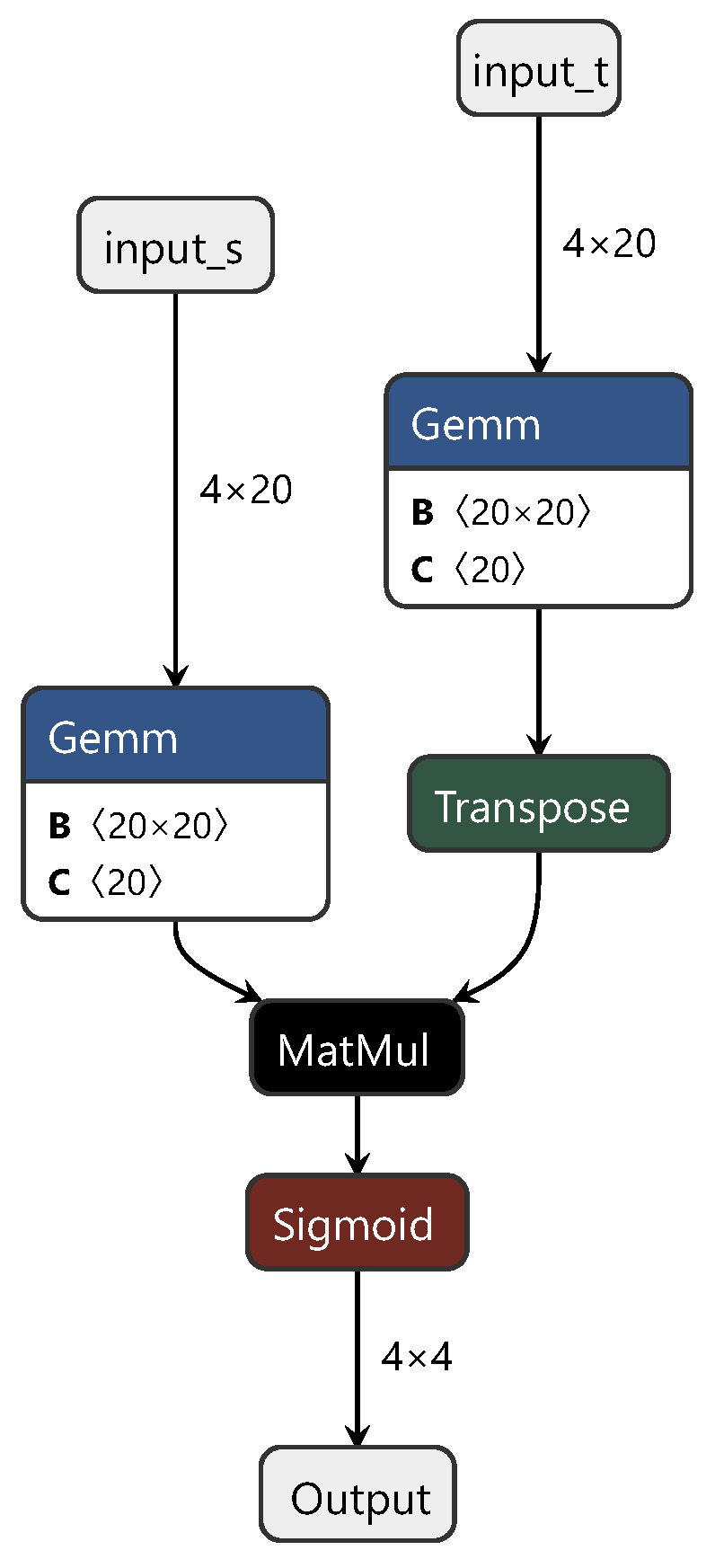
Diagram of the enhanced transport distance structure.

**Figure 6 sensors-24-01245-f006:**
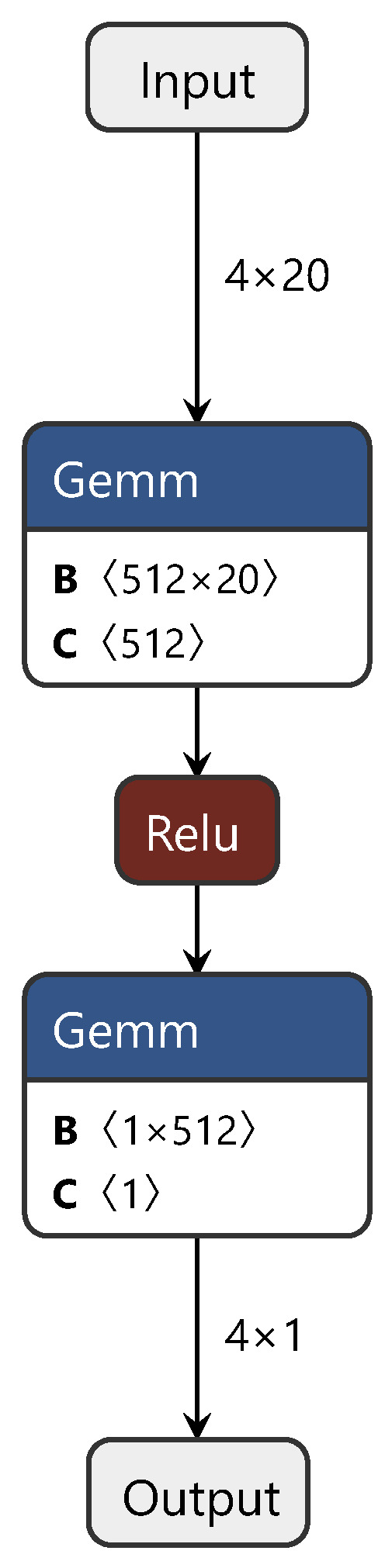
Diagram of the enhanced transport distance structure.

**Figure 7 sensors-24-01245-f007:**
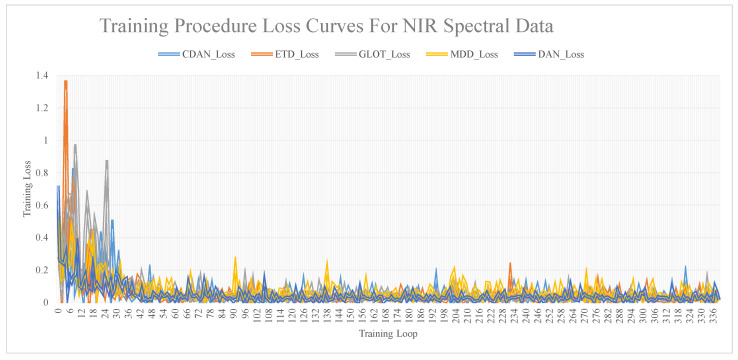
Training procedure loss curves for NIR spectral data.

**Figure 8 sensors-24-01245-f008:**
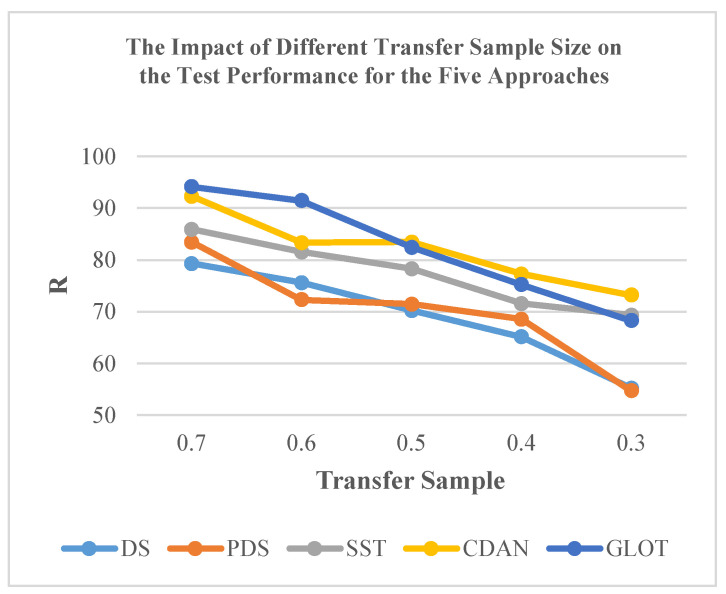
Impact of transfer sample size on the test performance of the five approaches.

**Figure 9 sensors-24-01245-f009:**
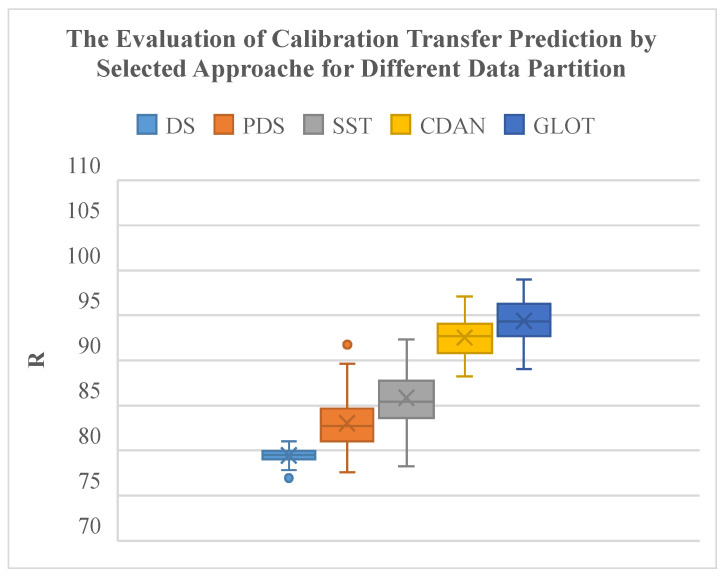
Evaluation of CT prediction of selected approaches for different data partitions.

**Table 1 sensors-24-01245-t001:** Characteristics of NIRS datasets.

NIRS Analyzer	Wavelength	Dataset Size
NIRquest512	512	196
NIR–NT–spectrometer–OEM system	118	270

**Table 2 sensors-24-01245-t002:** Model performance (R-squared, R) of different transfer networks in the target domain.

Training Dataset Split Ratio	CDAN	ETD	GLOT	MDD	DAN
0.7	92.33 ± 1.32	90.16 ± 1.81	94.16 ± 1.37	88.19 ± 1.49	87.57 ± 2.85
0.65	86.81 ± 2.27	89.34 ± 1.49	92.12 ± 1.48	87.73 ± 1.48	86.82 ± 1.81
0.6	83.33 ± 2.31	85.17 ± 2.28	91.47 ± 1.85	85.39 ± 2.34	83.83 ± 3.36
0.55	80.73 ± 3.17	84.19 ± 2.79	89.12 ± 2.45	84.38 ± 2.48	88.28 ± 2.84
0.5	83.45 ± 1.32	82.97 ± 2.29	82.45 ± 2.79	85.19 ± 2.17	82.49 ± 1.47
0.45	81.29 ± 2.51	79.63 ± 2.28	81.80 ± 2.57	82.27 ± 2.21	81.84 ± 2.85
0.4	77.28 ± 2.49	71.27 ± 2.01	75.23 ± 2.95	78.23 ± 2.39	80.19 ± 2.34
0.35	75.82 ± 4.92	72.72 ± 2.93	72.45 ± 2.71	76.90 ± 3.87	77.92 ± 2.48
0.3	73.19 ± 4.83	70.34 ± 3.34	68.27 ± 2.49	74.18 ± 3.41	74.82 ± 2.60
0.25	72.10 ± 4.31	72.23 ± 1.89	67.56 ± 2.45	75.54 ± 3.23	72.65 ± 2.77

**Table 3 sensors-24-01245-t003:** Performance of different CT methods in the target domain.

	DS	PDS	SST	CDAN	GLOT
R	79.29	83.46	85.91	92.33	94.16
RMSECV	15.169	13.756	12.809	7.359	10.692
R	70.23	72.8	76.53	82.11	83.59
RMSEP	23.967	19.395	21.798	12.237	11.582

## Data Availability

https://github.com/GrudgeChiang/NIR-Wood-Testing-Transfer-Redesign, accessed on 7 January 2024.

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
