# Peer review of "Efficient Near-Infrared Spectrum Detection in Nondestructive Wood Testing via Transfer Network Redesign"

_sensors, 2024, doi:10.3390/s24041245_

Round 1

Reviewer 1 Report

Comments and Suggestions for Authors

The article discusses the application of near-infrared spectroscopy (NIRs) in analyzing various samples, emphasizing its rapid, user-friendly, and nondestructive nature. 
Overall, this article is informative and points to exciting advancements in NIRs technology with the integration of deep learning.
The study focused on addressing the high costs and difficulties associated with obtaining annotated near-infrared (NIR) spectral data by employing five transfer learning methods. These methods are tested to determine their performance, particularly in terms of classification accuracy in scenarios with limited training data. The results show that the implemented transfer learning models can significantly reduce the need for extensively labeled NIR datasets due to their superior classification accuracy.
Minor revisions suggested for the artcle:
- Consider providing a brief overview of how each transfer learning method specifically contributes to the improved performance in classification accuracy, as this would help to substantiate the claims made.
- Provide a clearer distinction between the study's proposed methods and the prevalent CT methods, possibly by detailing the limitations of the latter that the former overcome.
- To enhance the readability, consider breaking down complex sentences into shorter, more digestible ones, especially when explaining the technical aspects of the neural network layers and domain adaptation algorithms.

Author Response

Q1: Consider providing a brief overview of how each transfer learning method specifically contributes to the improved performance in classification accuracy, as this would help to substantiate the claims made.

A1: Yes, in light of your feedback and that of the third reviewer, I have provided a detailed explanation in the Results and Discussion section regarding the challenging situation faced by deep network technology in establishing transfer calibration models for near-infrared spectra. Currently, we can only make limited judgments on the issue based on the model accuracy results.

Q2: Provide a clearer distinction between the study's proposed methods and the prevalent CT methods, possibly by detailing the limitations of the latter that the former overcome.

A2: Thank you for raising this point. I had similar thoughts before writing the paper, but I overlooked the necessity of demonstrating "the advanced nature of domain adaptation and the limitations of methods like PDS." I have now revised lines 619-635 of the paper to elaborate on the limitations of transfer networks and the PDS method.

Q3: To enhance the readability, consider breaking down complex sentences into shorter, more digestible ones, especially when explaining the technical aspects of the neural network layers and domain adaptation algorithms.

A3: To improve readability, I have submitted the revised manuscript to the editorial office for further editing. They will utilize their English editing service to refine the paper.

Reviewer 2 Report

Comments and Suggestions for Authors

The authors describe training of several neural network architectures on NIR-data from wood samples. Different NN models were implemented using the PyTorch software library and benchmarked against each other based on a loss function.

The paper has the character of a review paper on neural network technologies and only little information on the concrete research (classification of wood samples?) is given. The authors should elaborate a bit more, what the actual concrete goal of the described exercise was. Is the focus on classification of different wood types or identification of certain chemical functional groups typically found in wood or predicting spectra measured on instrument "A" using data from instrument "B"? This is not clear and should be explained.

The authors claim that neural network methods have an advantage over "classical" partial least squares regression approaches. Maybe it would be enlightening to demonstrate this superiority based on the current data set and include a classical approach for comparison.

In Figure 1 a legend would be helpful. How are the depicted peaks from IR spectra related to wood samples? What is the meaning of labels "(a)" - "(c)"?

In Figures 2 - 6 several symbols are not defined or explained, e.g. in Figure 2: What is the relevance of "input.1" and "input.6"? Why was a dimension of <20> selected? What is the meaning of "76" and "73" at the bottom of the figure? How are "Squeeze", "Unsqueeze", "slice" and "concat" operations defined? Since the introduction and review section of the paper are quite elaborate, some more words should be invested in describing the actual research.

For Figures 2-6 it would help to link the labels in the figures to the symbols in the equations introduced in section 2 of the paper.

Results section: Was any optimisation of the parameters of the optimiser performed? How do the authors justify to use the same parameters for all neural network models? Shouldn't there be different optimal settings for each network type?

Comments on the Quality of English Language

The English language is mostly fine, sometimes lab jargon or odd figures of speech have slipped in.

Author Response

Q1: The paper has the character of a review paper on neural network technologies and only little information on the concrete research (classification of wood samples?) is given. The authors should elaborate a bit more, what the actual concrete goal of the described exercise was. Is the focus on classification of different wood types or identification of certain chemical functional groups typically found in wood or predicting spectra measured on instrument "A" using data from instrument "B"? This is not clear and should be explained.

A1: Your concern is valid, and I have added a descriptive section about how the dataset was constructed. Reference values for tensile strength in solid wood panels samples were determined using official analysis methods. On solid wood panel samples, the tensile strength parallel to grain of wood (GB/T 1938.1-2009) was used to determine tensile strength, which was proposed by the Standardization Administration of China (SAC, 2003). In addition, I have clarified the specific goal of the study in the "Experimental Analysis of NIRs Datasets" section, highlighting the evaluation of five transfer network models developed: DAN, CDAN, MDD, ETD, and GLOT, aimed at surpassing the accuracy of traditional models such as PDS and DS.

Q2: The authors claim that neural network methods have an advantage over "classical" partial least squares regression approaches. Maybe it would be enlightening to demonstrate this superiority based on the current data set and include a classical approach for comparison.

A2: I apologize for not providing clear information on the NIR spectroscopy dataset used and the specific objectives of this research, which caused confusion. I have now supplemented the paper with a detailed description of the research goals in the introduction. This study focuses on developing transfer models, not calibration models, and compares them with PDS and DS calibration transfer models. Additionally, the assertion regarding the superiority of deep neural networks over PLS calibration models mentioned in the introduction is cited from other studies.

Q3: In Figure 1 a legend would be helpful. How are the depicted peaks from IR spectra related to wood samples? What is the meaning of labels "(a)" - "(c)"?

A3: Your constructive feedback is appreciated. I have added a description explaining the significance of labels "(a)" - "(c)" in Figure 1 and how they relate to wood samples and spectral peaks. Please refer to the following excerpt from the paper:

In Figure (a), it can be observed that the original signal of the NIR/IR spectrum comprises multiple absorption peaks denoted as $x$. The NIR/IR spectrum is viewed as a response $s$ formed by the "interweaving" of these original signals. The figure cited from Tooke's article illustrates how the interwoven NIR spectrum can be separated into multiple absorption peaks. The objective of deconvolution is to deduce the original signal $x$ from the response and impulse response $s". Figures (b) and (c) describe this process In Figure (c), a single absorption peak represents a characteristic functional group of the tested substance. For this study, this may correspond to C-H or O-H groups associated with cellulose.

Q4: In Figures 2 - 6 several symbols are not defined or explained, e.g. in Figure 2: What is the relevance of "input.1" and "input.6"? Why was a dimension of <20> selected? What is the meaning of "76" and "73" at the bottom of the figure? How are "Squeeze", "Unsqueeze", "slice" and "concat" operations defined? Since the introduction and review section of the paper are quite elaborate, some more words should be invested in describing the actual research.

A4: Thank you for your praise regarding the review section. You are correct; I neglected to define many parameters that are self-evident within the codebase. "input.1" represents target domain data, while "input.6" represents source domain data. The dimension of <20> was chosen for the extracted features layer from the fully connected layer. "76" and "73" represent the number of groups in the source and target domain data, respectively. "Squeeze", "Unsqueeze", "slice", and "concat" operations refer to slicing and concatenating operations used in processing torch data with PyTorch APIs. I have added these explanations to the figure captions.

Q5: For Figures 2-6 it would help to link the labels in the figures to the symbols in the equations introduced in section 2 of the paper.

A5: Most of the formulas discussed in my paper fall within the scope of backpropagation in neural networks and cannot be visually represented in the figures. The figures only depict the fully connected layers used.

Q6: Results section: Was any optimization of the parameters of the optimizer performed? How do the authors justify to use the same parameters for all neural network models? Shouldn't there be different optimal settings for each network type?

A6: Yes, I am unsure about how other researchers conducted their studies. It may be because I modified the aforementioned transfer models and did not use convolutional structures, or it could be because the PyTorch project maintenance team has already optimized the performance of various optimizers to the extreme. Based on my experimental results, the parameters of the optimizer hardly affect the final results of my model with increasing iterations, and changes in optimizer parameters even have less impact on the model results than the variability caused by randomness. The model results are almost entirely correlated with the amount of data and changes in network structure. Regarding optimizer parameters, I referred to the optimizer parameters published by the project team that proposed the CDAN model on GitHub.

Reviewer 3 Report

Comments and Suggestions for Authors

This research was focused on the systematic development of a deep transfer network for Near-Infrared spectrum detection, and aimed to enhance existing models, ensuring their practical applicability in the field of chemometrics. It seems to be an interesting study.

After evaluating the entire document, I have the following suggestions:

1 – Some numerical results should be added in the abstract.

2 – I suggest that the keywords be placed in alphabetical order.

3 – Between lines 35 and 36: I suggest including other recent applications of NIR spectrometer, including portable devices, such as:

- Portable Instruments Based on NIR Sensors and Multivariate Statistical Methods for a Semiautomatic Quality Control of Textiles. Machines 2023, 11, 564. https://doi.org/10.3390/machines11050564

- Characterization of crude oils with a portable NIR spectrometer. Microchemical Journal 181 (2022) 107696. https://doi.org/10.1016/j.microc.2022.107696

- Transfer learning strategy for plastic pollution detection in soil: Calibration transfer from high-throughput HSI system to NIR sensor. Chemosphere, Volume 272, June 2021, 129908. https://doi.org/10.1016/j.chemosphere.2021.129908

- A calibration friendly approach to identify drugs of abuse mixtures with a portable near-infrared analyzer. Drug Testing and Analysis, 2022, 14(6), 1089-1101. https://doi.org/10.1002/dta.3231

I understand that it can show Sensors readers how cross-sectional your study is.

4 – “Section 2.1. Fundamental Structure of Deep Neural Networks” can be reduced.

5 – There is an imbalance between section 2 (12 pages for Theories and Methods) against section 3 (only 4 pages for Results and discussion).

6 – Table 3: What was the variability measure that the authors used to accompany CDAN, ETD, GLOT, MDD, and DAN? Uncertainty measurement, confidence interval, standard deviation, etc…

7 – Did the authors assess whether the behaviour of the data is normal or Gaussian? And about the presence of outliers?

8 – Figure 9: Why did the authors use interquartile ranges (Boxplot) instead of confidence intervals?

9 – There have already been too many earlier studies with NIR outcomes utilizing PLSR, GA and SPA models, even though they are targeting distinct substances. What makes this study unique or different from others?

10 – As Raman is also used for chemometric approaches in the energy field, please highlight the benefits of NIR over Raman.

11 – I suggest that the Conclusions focus more on responding to the objectives. Furthermore, suggestions for future work should be included.

Comments on the Quality of English Language

Minor editing of English language required

Author Response

1 - Some numerical results should be added in the abstract.

A1: The abstract has been revised to include numerical results.

2 - I suggest that the keywords be placed in alphabetical order.

A2: Done.

3 - Between lines 35 and 36: I suggest including other recent applications of NIR spectrometer, including portable devices.

A3: The referenced literature has been included in the introduction.

4 - "Section 2.1. Fundamental Structure of Deep Neural Networks" can be reduced.

A4: Section 2.1 has been appropriately condensed.

5 - There is an imbalance between section 2 (12 pages for Theories and Methods) against section 3 (only 4 pages for Results and discussion).

A5: Section 3 has been expanded to 7 pages to address the imbalance.

6 - Table 3: What was the variability measure that the authors used to accompany CDAN, ETD, GLOT, MDD, and DAN?

7 - Did the authors assess whether the behaviour of the data is normal or Gaussian? And about the presence of outliers?

8 - Figure 9: Why did the authors use interquartile ranges (Boxplot) instead of confidence intervals?

A6-A8: Thank you for your questions. They indeed reflect common challenges encountered when considering deep neural networks in chemometrics. I have reworked some sections in the results and discussion, and I have provided detailed explanations of this issue from line 542 to line 591.

9 - There have already been too many earlier studies with NIR outcomes utilizing PLSR, GA and SPA models. What makes this study unique or different from others?

A9: The study's uniqueness lies in its comparison of transfer networks with traditional NIR calibration transfer methods. This has been detailed in lines 619-635.

10 - As Raman is also used for chemometric approaches in the energy field, please highlight the benefits of NIR over Raman.

A10: This study focuses on chemometrics, and the proposed models can be applied to both NIR and Raman spectroscopy, addressing calibration transfer challenges.

@article{LI2019143, title = {A calibration transfer methodology for Standardization of Raman instruments with different spectral resolutions using Double Digital Projection Slit}, journal = {Chemometrics and Intelligent Laboratory Systems}, volume = {191}, pages = {143-147}, year = {2019}, issn = {0169-7439}, doi = {https://doi.org/10.1016/j.chemolab.2019.07.004}, url = {https://www.sciencedirect.com/science/article/pii/S0169743919302667}, author = {Qifeng Li and Xueqing Sun and Xiangyun Ma and Bin Li and Huijie Wang and Haiyue Lv and Qing Wang and Kexin Xu and Da Chen} }

11 - I suggest that the Conclusions focus more on responding to the objectives. Furthermore, suggestions for future work should be included.

A11: Done.

Reviewer 4 Report

Comments and Suggestions for Authors

The authors studied the transfer learning methods for NIR spectral data processing. However, the title of the manuscript is very hard to understand. The main text is also not well-organized and clearly presented. 

Introduction. Why did the authors use so many single paragraphs? In addition, the challenges and problems in the current stage are not provided, which makes the motivation of study unclear.

The results are not clearly described. In Table 4, what did the letter 'R' at row 1 and row 3 mean? Table 3, what did the values in the table mean (what kind of performance indicators)?  Table 2, what is the purpose of listing the training loss here?

Due to the mentioned problems, I am sorry that in the current version it is very hard to read and understand the main idea of the study. Therefore, I suggest that the manuscript should be re-sumbitted after substantial revision. Please clearly describe the problems be addressed in this study and clearly state the innovative work or new discovery of this study.

Comments on the Quality of English Language

The English writting should be siginificantly improved. Some unclearly descriptions should be corrected. For example, what did 'Spectrum Detection' mean? I seems that 'wood quality/properties evaluation' or 'spectra analysis' can be the choice to describe the work in this manuscript. Similar problems can be found at the rest part of the manuscript.

Author Response

Q1: The authors studied the transfer learning methods for NIR spectral data processing. However, the title of the manuscript is very hard to understand. The main text is also not well-organized and clearly presented.

A1: Your feedback is valuable, and I appreciate your insight. I have revised the title of my manuscript to "Efficient Near-Infrared Spectrum Detection in Wood Non-Destructive Testing via Transfer Network Redesign". Additionally, I am currently reorganizing the introduction section.

Q2: Introduction. Why did the authors use so many single paragraphs? In addition, the challenges and problems in the current stage are not provided, which makes the motivation of the study unclear.

A2: Thank you for your comments on the introduction section. I have consolidated the single paragraphs into one coherent paragraph. Also, I have added the following paragraph at line XX (to be confirmed):

"However, similar to deep learning, the basic structures in domain adaptation methods have not been thoroughly analyzed from the perspectives of NIR spectroscopy mechanisms and chemometrics. To address this issue, this study aims to:

Analyze whether basic structures of deep networks, including convolutional structures, fully connected structures, and pooling structures, along with domain adaptation networks such as maximum mean discrepancies, Optimal Transport, and Adversarial Machine Learning, can be applied to NIR spectroscopy analysis techniques.
Based on our analysis, redesign five mainstream domain adaptation networks to adapt to the requirements of NIR spectroscopy calibration transfer models, facilitating their application in NIR spectroscopy analysis techniques."

Q3: The results are not clearly described. In Table 4, what did the letter 'R' at row 1 and row 3 mean? Table 3, what did the values in the table mean (what kind of performance indicators)? Table 2, what is the purpose of listing the training loss here?

A3: 'R' stands for the correlation coefficient. I have provided an explanation of this indicator at line 523. Table 3 also represents correlation coefficients, and I have included this information in the title of Table 3. The training loss listed in Table 2 serves as an initial reference indicator for evaluating the performance of these five models, and I have added this explanation to the description of Table 2.

Q4: Due to the mentioned problems, I am sorry that in the current version it is very hard to read and understand the main idea of the study. Therefore, I suggest that the manuscript should be re-submitted after substantial revision. Please clearly describe the problems to be addressed in this study and clearly state the innovative work or new discovery of this study.

A4: I have revised the abstract, title, introduction, and discussion and conclusion sections.

Round 2

Reviewer 2 Report

Comments and Suggestions for Authors

While the modification added some crucial context and made the manuscript more readable, I would still recommend to add more descriptions to figures 2-5. The authors mentioned modifications to figure captions in their response, but apparently these didn't make it into the new version of the manuscript. In the current version the figures are only understandable in combination with a manual for the PyTorch API. The editor may correct me if I'm wrong, but not all reads of the journal can be expected to have such deep knowledge of this software library. Sorry for nagging about this... 

Comments on the Quality of English Language

Isn't this section redundant? What is the purpose of the multiple choice menu about English language above? 

Author Response

Q1: While the modification added some crucial context and made the manuscript more readable, I would still recommend adding more descriptions to figures 2-5. The authors mentioned modifications to figure captions in their response, but apparently these didn't make it into the new version of the manuscript. In the current version, the figures are only understandable in combination with a manual for the PyTorch API. The editor may correct me if I'm wrong, but not all readers of the journal can be expected to have such deep knowledge of this software library. Sorry for nagging about this...

A: Your suggested modifications are entirely correct. Due to the short turnaround time for revisions of the previous draft, I didn't manage to incorporate all the changes. However, I have now updated the input and output modules from Figure 2 to Figure 5 and provided some explanations for these figures. For example:

Figure 2: In Figure \ref{fig2}, "input_t" and "output_t" represent target domain data, while "input_s" and "output_s" represent source domain data. "Squeeze," "Unsqueeze," "slice," and "concat" correspond to the PyTorch API instructions used for slicing and concatenation when handling torch data. Currently at line 360.

Figure 3: In this figure, "gemm" denotes the matrix multiplication between spectral features and weights, while "relu" and "sigmoid" represent activation functions. This figure depicts the branch network. Currently at line 429.

Figure 4: To highlight the improvements achieved with the MDD network, only the structure diagram of the minimax optimization adversarial layer is shown in Figure \ref{fig4}. In this diagram, "sub" represents matrix subtraction. Currently at line 455.

Figure 5: Figure \ref{fig5} shows the network structure of the Kantorovich potential layer in the ETD. In this diagram, "matmul" represents the matrix multiplication between the spectral feature tensor and the weight tensor, while "transpose" denotes matrix transposition. Currently at line 489.

Regarding the issue with the PyTorch API, at present, I can only partially indicate in the manuscript that these modules are the result of PyTorch API functions operating on arrays.

Reviewer 3 Report

Comments and Suggestions for Authors

Author Response

A1: The abstract has been revised to include numerical results.
I suggest not using these abbreviations in the abstract without previous definition: CDAN, GLOT, DS, PDS, SST and RMSEP.
A: yes, we done. 

2 I suggest that the keywords be placed in alphabetical order.
A2: Done.
Please, check it again!
A: Apologies, I forgot to remove the middleware of the LaTeX program, and I didn't completely re-run and generate the PDF file. 

A10: This study focuses on chemometrics, and the proposed models can be applied to both NIR
and Raman spectroscopy, addressing calibration transfer challenges.
I suggest including this information in the manuscript. Thank you!
A: Yes, I have added this information to line 46 of the manuscript. 

Reviewer 4 Report

Comments and Suggestions for Authors

The authors have improved the manuscript. There are some problems found in the current version. 

As I mentioned in the first round of review (comment #3), 'In Table 4, what did the letter 'R' at row 1 and row 3 mean? Table 3, what did the values in the table mean (what kind of performance indicators)? Table 2, what is the purpose of listing the training loss here? ', In Table 4, what did the letter 'R' at row 1 and row 3 mean? I guess one is for crossvalidation and another is for prediction? Please distinguish them using subscripts or suffixes. Then, at row 2 and row 4, the definition of RMSEP can be found in the Abbreviations section, but the definition of RMSECV was missing. Please carefully check the rest part of the manuscript again and correct such problems.

Table 2, what is the purpose of listing the training loss here? Tables are usually used to summarize and show some important data and results. As a fact, the training loss cannot explain any problem. In theory, as long as the model is complex enough, the training loss can be infinitely reduced. I suggest that just explain the convergence of the model here.

Comments on the Quality of English Language

none

Author Response

Q1: As I mentioned in the first round of review (comment #3), "In Table 4, what does the letter 'R' at row 1 and row 3 mean? Table 3, what do the values in the table represent (what kind of performance indicators)? Table 2, what is the purpose of listing the training loss here?" In Table 4, what does the letter 'R' at row 1 and row 3 mean? I guess one is for cross-validation and another is for prediction? Please distinguish them using subscripts or suffixes.

A1: I apologize for the rush in addressing the issues in the previous round of revisions. Some concerns were not properly addressed. 'R' represents the correlation coefficient, as mentioned in line 588 (Due to the focus of deep networks on inputs and outputs, researchers have predominantly used metrics such accuracy, F1 score, coefficient of determination (R-squared, R)). The correlation coefficient is commonly denoted as R, R^2, or R-squared. In this study, we chose 'R' to represent this parameter.

Q2: Then, at row 2 and row 4, the definition of RMSEP can be found in the Abbreviations section, but the definition of RMSECV was missing. Please carefully check the rest of the manuscript again and correct such problems.

A2: Some content was inadvertently omitted in the previous round of revisions. I have now made a correction in line 591:

Replacing the text in the template: "root mean square error (RMSE), root mean square error of prediction (RMSEP), etc., to interpret model quality. For our purposes, we set aside these controversies and chose R, RMSE, and RMSEP as metrics to measure the performance of the five models."

Modified to: "root mean square error (RMSE), Root Mean Square Error of Cross Validation (RMSECV), root mean square error of prediction (RMSEP), etc., to interpret model quality. For our purposes, we set aside these controversies and chose R, RMSE, RMSECV, and RMSEP as metrics to measure the performance of the five models."

Q3: Table 2, what is the purpose of listing the training loss here? Tables are usually used to summarize and present important data and results. As a matter of fact, the training loss cannot explain any problem. In theory, as long as the model is complex enough, the training loss can be infinitely reduced. I suggest explaining just the convergence of the model here.

A3: Yes, you are correct. I have removed Table 2.